# PET metabolic tumor volume as a new prognostic factor in childhood rhabdomyosarcoma

Helio Fayolle[1]*, Nina Jehanno[2‡], Valerie Lauwers-Cances[3], Marie-Pierre Castex[4‡], Daniel Orbach[5‡], Thomas Mognetti[6‡], Corradini Nadège[7‡], Pierre Payoux[8], Anne Hitzel[1]

1 Nuclear Medicine Department, Toulouse Purpan University Hospital, Toulouse, France, 2 Nuclear Medicine Department, Curie Institute, PSL Research University, Paris, France, 3 Epidemiology and Public Health Department, Faculty of Medicine, Toulouse University Hospital, Toulouse, France, 4 Paediatric Haemato-Oncology Department, Toulouse Children's Hospital, Toulouse University Hospital, Toulouse, France, 5 IREDO Oncology Centre, Curie Institute, PSL University, Paris, France, 6 Nuclear Medicine Department, Léon Bérard Cancer Centre, Lyon, France, 7 Oncology and Clinical Research Departments, Léon Bérard Cancer Centre and Institute of Paediatric Haematology and Oncology, Lyon, France, 8 Toulouse NeuroImaging Centre, Toulouse Paul Sabatier University-INSERM, Toulouse, France

☯ These authors contributed equally to this work.
‡ NJ, MPC, DO, TM and CN also contributed equally to this work.
* heliof@hotmail.fr

**Data Availability Statement:** All relevant data are within the paper and its Supporting Information files.

## Abstract

### Purpose

Childhood RMS is a rare malignant disease in which evaluation of tumour spread at diagnosis is essential for therapeutic management. F-18 FDG-PET imaging is currently used for initial RMS disease staging.

### Materials and methods

This multicentre retrospective study in six French university hospitals was designed to analyse the prognostic accuracy of MTV at diagnosis for patients with RMS between 1 January 2007 and 31 October 2017, for overall (OS) and progression-free survival (PFS). MTV was defined as the sum of the primitive tumour and the largest metastasis, where relevant, with a 40% threshold of the primary tumour SUVmax. Additional aims were to define the prognostic value of SUVmax, SUVpeak, and bone lysis at diagnosis.

### Results

Participants were 101 patients with a median age of 7.4 years (IQR [4.0-12.5], 62 boys), with localized disease (35 cases), regional nodal spread (43 cases), or distant metastases (23). 44 patients had alveolar subtypes. In a univariate analysis, a MTV greater than 200 cm$^3$ was associated with OS (HR = 3.47 [1.79;6.74], $p<0.001$) and PFS (HR = 3.03 [1.51;6.07], $p = 0.002$). SUVmax, SUVpeak, and bone lysis also influenced OS (respectively $p = 0.005$, $p = 0.004$ and $p = 0.007$) and PFS ($p = 0.029$, $p = 0.019$ and $p = 0.015$). In a multivariate analysis, a MTV greater than 200 cm$^3$ was associated with OS (HR = 2.642

**Funding:** The authors received no specific funding for this work.

**Competing interests:** The authors have declared that no competing interests exist.

**Abbreviations:** MTV, metabolic tumour volume; OS, overall survival; PFS, progression-free survival; RMS, rhabdomyosarcoma; PET, $^{18}$FDG-PET-CT; HR, Hazard Ratio; SUV, standardized uptake value; eRMS, embryonal rhabdomyosarcoma; aRMS, alveolar rhabdomyosarcoma; CNIL, national information science and liberties commission (Commission Nationale de l'Informatique et des Libertés in french).

[1.272;5.486], $p = 0.009$) and PFS (HR = 2.707 [1.322;5.547], $p = 0.006$) after adjustment for confounding factors, including SUVmax, SUVpeak, and bone lysis.

## Conclusion

A metabolic tumor volume greater than 200 cm$^3$, SUVmax, SUVpeak, and bone lysis in the pre-treatment assessment were unfavourable for outcome.

## Introduction

Soft-tissue sarcoma is the third most common tumour in the 0-19 years age group after blood diseases (lymphoma and leukaemia) and nervous system tumours [1]. It represents about 7% of cancers in children, while rhabdomyosarcoma (RMS) accounts for about half the number of childhood cases of soft-tissue sarcoma [2]. It is classically divided into two main histological types: alveolar, which is the most aggressive and makes up 15-20% of cases, and all nonalveolar subtypes [3].

After initial clinical symptoms, such as a swelling, paraclinical tests are used to confirm the diagnosis and to guide therapeutic management: lesion biopsy, local imaging (lung-CT, MRI), lumbar puncture in the case of a parameningeal primary, bone marrow aspiration, and biopsy [4]. MRI evaluates the local spread and extent of the disease before surgery, and CT assesses outside invasion, such as lymph node or organ metastatic lesions [5].

Identification of unfavourable factors such as age ($> 10$ years), alveolar histological subtype, size ($> 5$ cm) and location of the primary tumour (parameningeal, limbs, or trunk), and presence of regional nodal or distant metastatic tumour spread sites, has allowed a risk classification to optimize the treatment, leading to an improvement of the cure rate from 25–30% to approximately 70% [6]. Thus patients in the low risk group have the best prognosis progression-free survival (PFS) and overall survival (OS) (3-year PFS rate of 88%) [7]. However, patients with metastatic disease still have a dismal prognosis (OS: 0-30%) [8].

Apart from detecting intrathoracic lesions where chest CT remains essential, F-18 FDG PET-CT (PET) has been shown to be superior to CT for the initial staging of RMS, mainly on account of its ability to detect nodal involvement and metastatic disease [9]. Recent studies, albeit retrospective and with a limited number of patients, have reported equivocal results about the prognostic significance of metabolic tumour activity measured via standardized uptake value (SUV) [10–14].

Single pixel values of the SUV, and especially SUVmax, are commonly used as a quantitative index of tumor metabolism, mainly because it is now well implemented on images viewers and thus easy to use. However this semiquantitative evaluation is subject to intra- and interindividual biases by a broad range of biological and technical factors such as patient's weight, blood glucose level, acquisition parameters including uptake time, inaccurate calibration of PET, image reconstruction algorithm, etc. [15, 16]. To overcome these shortcomings, MTV approach, defined as the sum of the volume of voxels with SUV surpassing a threshold value in a tumor, can be considered [17]. Recent studies confirmed the interest of MTV and sometimes its superiority compared to SUVmax with regard to prognostic value, in various neoplastic pathologies such as Hodgkin's lymphoma, ovarian squamous cell carcinoma, non-small cell lung cancer, metastatic colorectal cancer, and pancreatic cancer [18–23].

The main aim of the present study was to assess the prognostic value of MTV measured on PET at diagnosis in children with rhabdomyosarcoma.

## Methods and methods

### Patient population

All children and adolescents aged between 2 months and 20 years who had undergone a PET-CT at diagnosis as part of the RMS work-up, and who were treated in the paediatric oncology departments of four French university hospitals (Clermont-Ferrand, Marseille, Montpellier and Toulouse) and two cancer centres (Paediatric Haemato-Oncology Institute in Lyon and Curie Institute in Paris) within the framework of the EpSSG-RMS-2005 European protocol, Bernie protocol or other randomised controlled, Phase 3 trials, between 1 January 2007 and 31 October 2017 [24, 25], were selected.

According to the french law, the retrospectives non interventional studies do not require patient consent when the study protocol is compliant with the CNIL reference methodology repository about the retrospective data collection. In the case of an expressly written refusal, patient data were not analysed. This has allowed our work to be approved by the Ethics Committee of French Society of Nuclear Medicine and registered with the number CEMEN 2020–01. Thus, we could analyse PET images from patients and provide illustrative images.

Patients who had undergone a PET-CT examination while the primary tumor was already excised by surgery, who had commenced chemotherapy, who had RMS located in the bladder or in a parameningeal site (where measuring MTV was impossible, owing to the close physiological activity of the bladder or brain), whom PET images where unrecorded, or who had a tumour in a limb that was not within the scope of acquisition, were excluded.

Of the 326 patients with histologically confirmed RMS, 101 were selected (Fig 1).

Information about the history of the disease was retrospectively collected, notably the stage determined during the pretreatment clinical assessment of disease, the COG-STS risk stratification, prior resection of the primitive tumour, histological subtype and stage. The RMS stage (ranged from 1 to 4) depended on the anatomical site of the primary tumour, tumour size (above or below 5 cm), presence of regional lymph node involvement, presence of distant metastasis [26].

### Treatment

All patients were included or treated as per the EPSSG-RMS-2005 and Bernie protocols according to their risk group, combining surgery, chemotherapy and radiotherapy. 39 of the

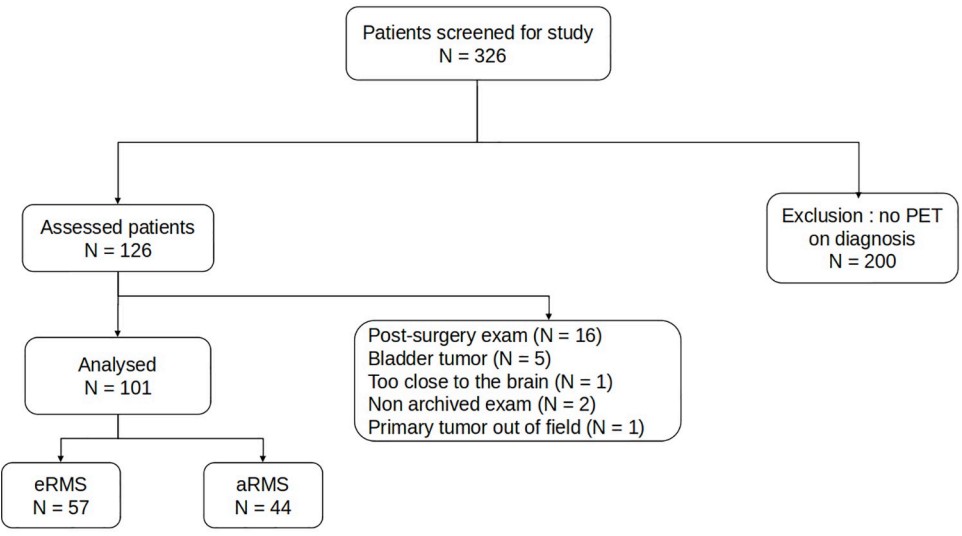

**Fig 1. Flowchart.** N: number of patients.

101 patients have been previously reported [24, 25, 27]. This prior article dealt with prognosis factors especially the MTV whereas previous studies compared the efficacy of different treatment exposure. The monitoring methods after the end of the treatment were also defined by the protocols. All children and adolescents aged 0-20 years who had been diagnosed with localized or metastatic RMS were prospectively included. Histological diagnosis was carried out by the local pathologist and reviewed by the EpSSG Pathology Panel. Alveolar subtype was based on histology, as fusion status was not mandatory. Each tumour was classified according to its site of origin. Evaluation of lymph node involvement was primarily based on MRI or CT, but was verified by sampling on suspicion. Regional lymph nodes were defined as those appropriate to the site of the primary tumour. Induction chemotherapy was administered according to the EpSSG risk group, with liver-bile duct RMS considered an unfavourable *other site*. Patients within high-risk group were first randomized to receive either ifosfamide, vincristine and actinomycin (IVA regimen) or IVA + doxorubicin (IVADo regimen). Those who achieved complete remission (CR) after nine courses cycles of induction chemotherapy and local therapy entered a second randomization phase, receiving either no further chemotherapy, or 6 months of maintenance chemotherapy using vinorelbine and cyclophosphamide (VC). Local treatment was recommended after four cycles of induction chemotherapy, and was decided by the local multidisciplinary team, using delayed surgery, radiotherapy, or both. Patients with metastatic disease received induction chemotherapy (four cycles of IVADo + five cycles of IVA, ± bevacizumab), surgery and/or radiotherapy, followed by maintenance chemotherapy (12 cycles of low-dose cyclophosphamide + vinorelbine). Local therapy (surgery + radiotherapy) was planned after six courses.

## PET protocol

All patients fasted for at least 5 hours prior to injection and had a blood sugar level below 120mg/dl. The activity of the FDG administered intravenously 60 minutes before examination was adapted to the patients' weight and age, in accordance with the EANM paediatric calculators (https://www.eanm.org/publications/dosage-calculator/).

The following PET equipment was used in the six sites: in Clermont-Ferrand, a General Electric Discovery ST and a General Electric Discovery ST710, in Lyon, a Philips Gemini Allegro Body then, from 2012 onwards, a Philips Gemini Big Bore; in Marseille, a GE discovery until 2010, then a Siemens biograph 16; in Montpellier, a Siemens Biograph until 2015, then a Siemens mCT20 flow; in Toulouse, a Siemens biograph 6.0 Truepoint Hirez; and in Paris, a Philips Gemini XLI until 2016, then a General Electric Discovery ST710.

The PET-CT scans were interpreted by two nuclear medicine physicians, of 2 and 16 years of experience, blinded to the clinical data except for the pathological diagnosis. MTVs were measured using the same imaging analysis software (TrueD-Siemens SyngoCT2006A). Studies were read independently, and in case of disagreements the exams were interpreted once again with the two physician to find a consensus agreement.

An initial visual analysis was performed to determine the sites of abnormal FDG uptake. Any uptake greater than the adjacent background activity and which could not be explained by physiological or inflammatory phenomena was considered to be pathological.

SUVmax, SUVpeak and MTV were calculated by placing a spheroid-shaped volume of interest at the site of the primary tumour lesion.

From the physics side, after phantom studies [28, 29], a threshold value of 40% to define the tumor boundary on PET images was used in many clinical studies [30–33]. This 40% threshold is the most common index in clinical practice for evaluating tumor prognosis [34, 35]. Nowadays PET imaging softwares offer an automatic 40% SUV approach to delineate tumor contours. Thereby a threshold of 40% of the SUVmax was applied in our study.

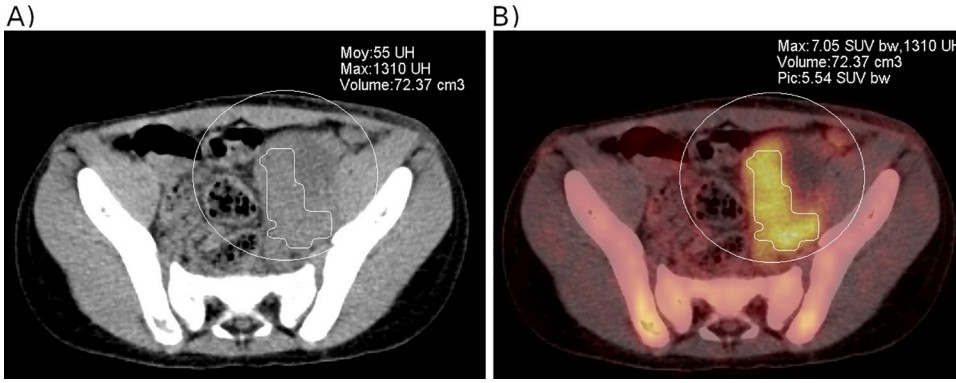

**Fig 2. Lumbar-aortic lymph node involvement in a 7-year-old child with a pelvic alveolar RMS.** Metabolic tumour volume represents all the voxels within the initial volume of interest with SUV values equal to or above an automatic threshold of 40% of SUVmax. A) CT scan: the volume is hardly measurable, mainly owing to the presence of extensive necrosis. B) Fused PET-CT with metabolic tumour volume, SUVmax and SUVpeak.

Tumour volume was delineated by all the voxels within the initial volume of interest, with SUV values equal to or above this threshold (Fig 2).

In the event of nonlocalized disease, we deemed that exhaustive measurement of all the lesions was impossible in daily clinical practice. We therefore adopted an approach whereby the total tumour volume was measured as the sum of the primary tumour volume and the volume of the largest distant lesion (lymph node or distant metastasis). MTV therefore referred either to the metabolic volume of the primary tumour, or to the sum of the primary tumour and the largest distant lesion.

The presence of bone lysis induced by tumour contiguity on the PET-CT scan was also noted.

## Statistical analysis

$Chi^2$ or Fisher tests were used to compare categorical variables, and Student or Wilcoxon-Mann-Whitney tests to compare quantitative variables. PFS was defined as the time interval between diagnosis and progression or death, whichever occurred first. OS was defined as the time from diagnosis to death. PFS and OS were analysed using the Kaplan-Meier method, log-rank test, and Cox proportional hazards model to estimate the hazard ratio (HR), along with 95% confidence intervals (95% CIs). Follow-up was estimated using the reverse Kaplan-Meier method [36]. To test the prognostic value of MTV, a minimum $p$ value approach was used to determine the cut-off value [37]. In a multivariate analysis, the first model included all the possible confounding factors with a $p$ value $< 0.2$ in the univariate analysis. The model was reduced using backward elimination until only significant effects remained. First-order interactions were explored and proportionality checked. Tests were two-sided, and $p$ values below 0.05 were considered significant. Analyses were performed using STATA version 14.2 software (StataCorp).

## Results

Of the 326 patients with histologically confirmed RMS, 101 were eligible according to the pre-specified exclusion criteria (Fig 1). Patients' characteristics are shown in Table 1. Median age at diagnosis was 7.4 years (IQR [4.0-12.5], children with alveolar RMS were older ($p = 0.001$), and 61% were male. The most frequent histology subtype was nonalveolar RMS (56%). According to

Table 1. Patients' characteristics.

| Patients' characteristics | Overall | Alveolar subtype | Embryonal subtype | P value |
|---|---|---|---|---|
| N | 101 | 44 | 57 | |
| Age at diagnosis | | | | |
| Median [IQR] | 7.4 [4.0–12.5] | 10 [5.6–14.6] | 5.4 [3.5–10.1] | 0.001 |
| Sex (%) | | | | 1 |
| Male | 62 (61.4) | 27 (61.4) | 35 (61.4) | |
| Female | 39 (38.6) | 17 (38.6) | 22 (38.6) | |
| Stage (%) | | | | 0.007 |
| 1 | 18 (17.8) | 5 (11.4) | 13 (22.8) | |
| 2 | 17 (16.8) | 5 (11.4) | 12 (21.1) | |
| 3 | 43 (42.6) | 17 (38.6) | 26 (45.6) | |
| 4 | 23 (22.8) | 17 (38.6) | 6 (10.5) | |
| Primary tumour location (%) | | | | <0.001 |
| Extremity | 26 (25.7) | 19 (43.2) | 7 (12.3) | |
| Orbit, H/N (non-PM) | 21 (20.8) | 9 (20.5) | 12 (21.1) | |
| PM | 21 (20.8) | 4 (9.1) | 17 (29.8) | |
| Abdomen/pelvis (including genital and BP) | 24 (23.8) | 9 (20.5) | 15 (26.3) | |
| Paratesticular | 6 (5.9) | 0 (0.0) | 6 (10.5) | |
| Bone | 3 (3.0) | 3 (6.8) | 0 (0.0) | |
| Distant lesions (%) | | | | 0.008 |
| No | 63 (63.7) | 21 (47.7) | 43 (75.4) | |
| Yes | 37 (36.6) | 23 (52.3) | 14 (24.6) | |
| Distant lesion location | | | | 0.866 |
| Lymph node | 25 (67.6) | 16 (69.6) | 9 (64.3) | |
| Pulmonary | 4 (10.8) | 2 (8.7) | 2 (14.3) | |
| Bone | 8 (21.6) | 5 (21.7) | 3 (21.4) | |
| Bone lysis (%) | | | | 0.983 |
| No | 76 (76.2) | 33 (75) | 44 (76.2) | |
| Yes | 24 (23.8) | 11 (25) | 13 (23.8) | |
| Local surgery (%) | | | | 0.008 |
| No | 77 (76.2) | 21 (47.7) | 43 (75.4) | |
| Yes | 24 (23.8) | 23 (52.3) | 14 (24.6) | |

BP, bladder/prostate; Ext, extremity;; H/N, head and neck; PM, parameningeal.

RMS staging, 18 patients were classified as Stage 1, 17 as Stage 2, 43 as Stage 3 and 23 as Stage 4. 37% of patients had a distant site with lymph node involvement and 24 had a bone lysis induced by tumour contiguity, as this 4-year-old child with a left mandibular embryonal RMS (Fig 3). There were more patients with advanced stages in alveolar RMS ($p = 0.007$).

The PET characteristics are shown in Table 2. The median MTV of the primary tumour was 26 cm$^3$ (IQR [10.2-103], and the median MTV of the primary tumour plus largest distant lesion where relevant was 31 cm$^3$ (IQR [13.1-172]. MTV was used for prognosis in both OS and PFS, and because cut-off values are easier for clinicians to grasp and to classify (e.g. RMS staging in four steps), we calculated the best cut-off MTV, using the minimum $p$ value approach. An MTV of 200 cm3 was the optimum cutoff point in OS and PFS analyses, and 22% of patients had an MTV above 200 cm$^3$.

Median follow-up time for the cohort was 40 months IQR [23–64], 43 patients (42.6%) had disease progression, and 37 patients died (36.6%).during this period.

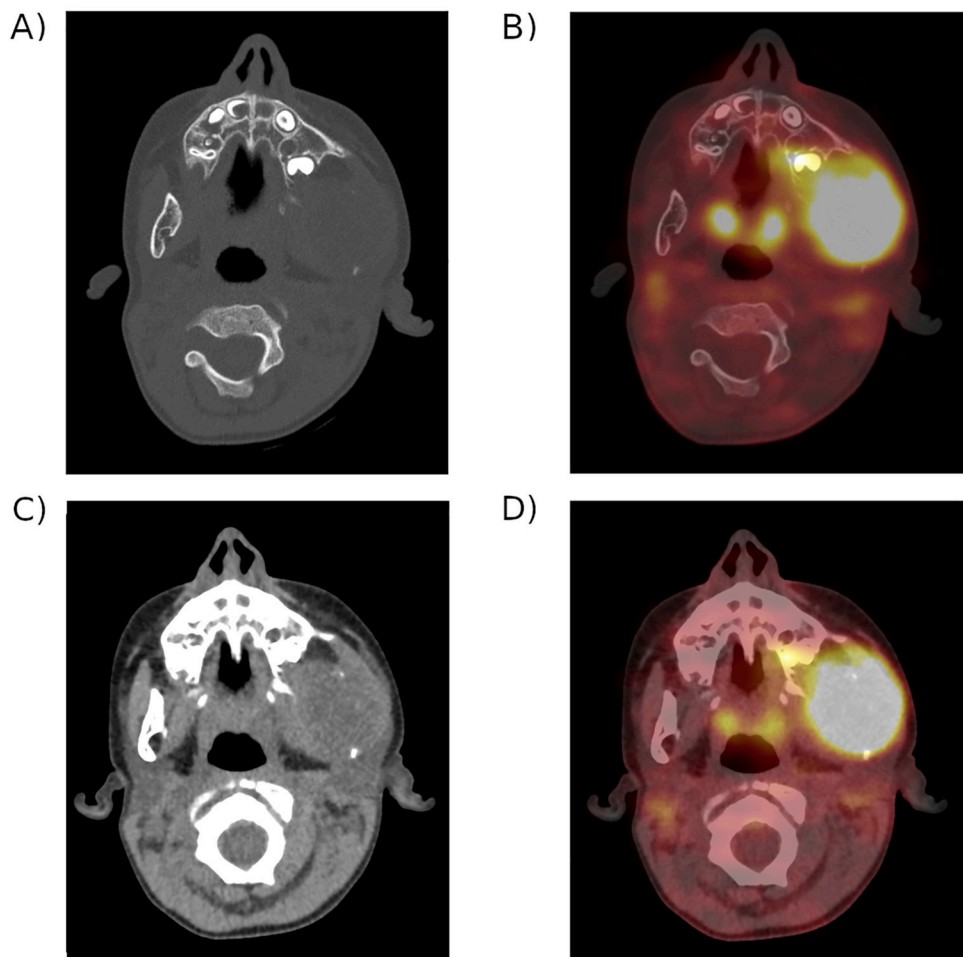

**Fig 3. Left mandibular isolated primary tumour in a 4-year-old child with embryonal RMS.** Primary tumour lysis of the left mounting mandibular branch. A) CT scan only in bone window. B) Fused PET-CT in bone window. C) CT scan only in soft-tissue window. D) Fused PET-CT in soft-tissue window.

**Table 2. PET characteristics.**

| | | | | |
|---|---|---|---|---|
| SUVmax of primary tumor median [IQR] | 5.5 [3.9; 7.6] | 5.8 [4.7; 7.6] | 5.5 [3.9; 7.6] | 0.289 |
| SUVpeak of primary tumour median [IQR] | 4.4 [3.4; 6.3] | 4.4 [3.4; 6.3] | 4.1 [2.6; 5.6] | 0.225 |
| Volume of primary tumour $cm^3$ median [IQR] | 26 [10.2; 103] | 30 [15.9; 128.8] | 20 [9.2; 54.3] | 0.207 |
| SUVmax of distant lesion median [IQR] | 4.6 [3.6; 7.1] | 4.6 [3.6; 6.6] | 4.8 [3.2; 9.1] | 0.283 |
| SUVpeak of distant lesion median [IQR] | 4.2 [2.7; 5.8] | 4.2 [2.7; 5] | 4.1 [2.6; 6.6] | 0.317 |
| Volume of distant lesion $cm^3$ median [IQR] | 27.1 [3.9; 81] | 58 [5.8; 83.5] | 21 [3.4; 65.4] | 0.719 |
| Volume > 200 $cm^3$ (%) (primor tumour only) | | | | |
| No | 85 (84.2) | 35 (79.5) | 50 (87.7) | - |
| Yes | 16 (15.8) | 9 (20.5) | 7 (12.3) | 0.400 |
| MTV > 200 $cm^3$ (%) (primor tumour + distant lesion) | | | | |
| No | 79 (78.2) | 30 (68.2) | 49 (86) | - |
| Yes | 22 (21.8) | 14 (31.8) | 8 (14) | 0.057 |

## Overall Survival (OS)

Median OS was 72.5 months (IQR [20.5-not reached] and the probability of surviving for 3 years after diagnosis was 62% (Fig 4A). Age at diagnosis, Stage 4, bone lysis induced by primary tumour contiguity, SUVpeak, SUVmax and an MTV above 200 cm$^3$ were prognostic

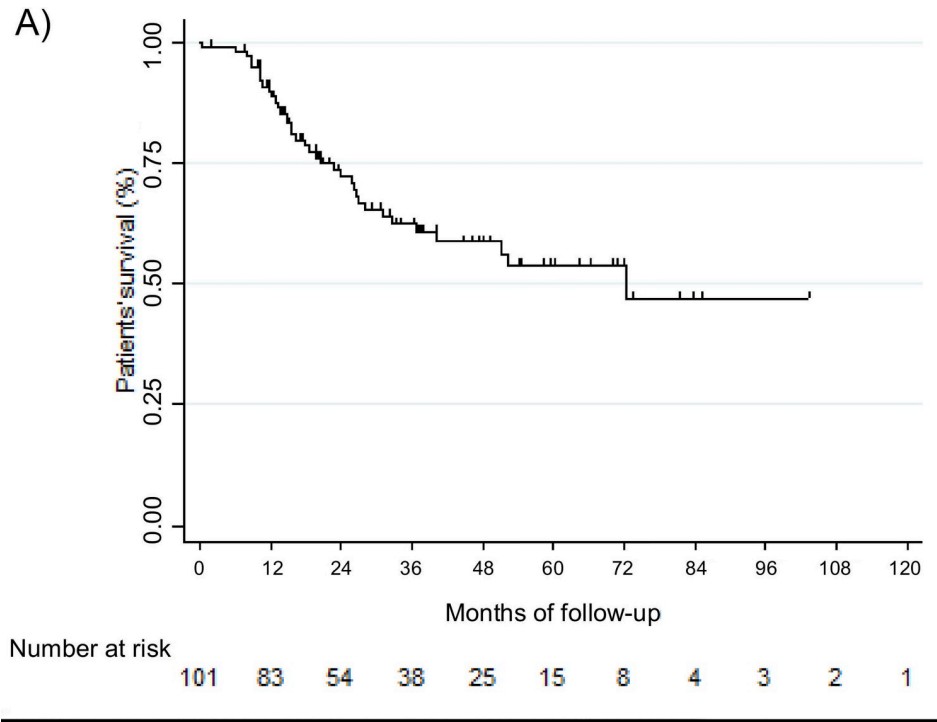

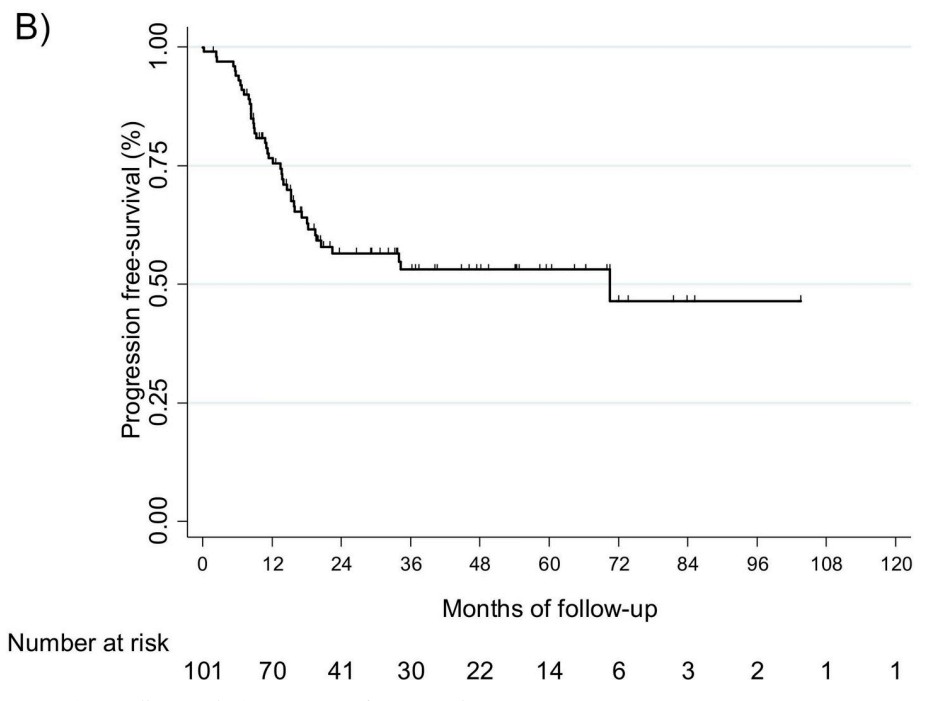

**Fig 4.** A) Overall survival. B) Progression-free survival.

**Table 3. Univariate Cox analysis, overall & progression-free survival.**

| Patients' characteristics | Overall survival HR [95% CI] | P value | Progression-free survival HR [95% CI] | *P* value |
|---|---|---|---|---|
| Age at diagnosis Median [IQR] | 1.10 [1.04–1.17] | 0.001 | 1.08 [1.02–1.14] | 0.006 |
| Sex | | | | |
| Male | 1 | - | 1 | - |
| Female | 0.88 [0.45;1.71] | 0.7 | 0.91 [0.49;1.70] | 0.781 |
| Histology | | | | |
| embryonal | 1 | - | 1 | - |
| alveolar | 0.69 [0.36;1.32] | 0.265 | 0.57 [0.31;1.05] | 0.071 |
| Stage | | | | - |
| 1 | 1 | - | 1 | |
| 2 | 0.76 [0.14;4.16] | 0.755 | 0.39 [0.08;1.91] | 0.243 |
| 3 | 2.24 [0.74;6.79] | 0.154 | 1.46 [0.58;3.68] | 0.577 |
| 4 | 6.19 [2.05;18.70] | 0.001 | 3.74 [1.47;9.53] | 0.006 |
| Bone lysis | | | | |
| No | - | - | 1 | - |
| Yes | 2.57 [1.29;5.10] | 0.007 | 2.23 [1.17;4.24] | 0.015 |
| Local surgery | | | | |
| No | 1 | - | 1 | - |
| Yes | 0.51 [0.27;0.98] | 0.041 | 0.56 [0.31;1.03] | 0.062 |
| SUVmax median [IQR] | 1.14 [1.04–1.24] | 0.005 | 1.10 [1.01–1.20] | 0.029 |
| SUVpeak median [IQR] | 1.15 [1.05–1.27] | 0.004 | 1.12 [1.02–1.24] | 0.019 |
| Volume (cm$^3$) (primor tumour only) | | | | |
| [0-200] | 1 | - | 1 | - |
| ]200+] | 3.81 [1.85;7.85] | <0.001 | 3.27 [1.44;7.46] | 0.005 |
| MTV cm$^3$ (primor tumour + distant lesion) | | | | |
| [0-200] | 1 | - | 1 | - |
| ]200+] | 3.47 [1.79;6.74] | <0.001 | 3.03 [1.51;6.07] | 0.002 |

factors of OS in the univariate analysis (Table 3). Patients with a metabolic primary tumour volume above 200 cm$^3$ had poorer OS (3.81 [1.85; 7.85]; $p < 0.001$) Patients with an MTV above 200 cm$^3$ also had a poorer OS than those who had an MTV equal to or below 200 cm3 (HR = 3.5; 95% CI [1.8; 6.7]; $p < 0.01$) (Fig 5)). In the multivariate analysis, after controlling for confounding effects, the statistically independent clinical or biological factors for OS that worsened patients' prognosis were Stage 4 ($p = 0.014$), bone lysis induced by primary tumour contiguity ($p = 0.003$), and MTV above 200 cm$^3$ ($p = 0.009$). Compared with patients who had an MTV equal to or below 200 cm$^3$, patients with an MTV above 200 cm$^3$ were 2.6 times more likely to die (adjusted HR = 2.6; 95% CI [1.3; 5.5] (Table 4). No statistically independent effect on OS was found for either SUVpeak ($p = 0.844$) or SUVmax ($p = 0.842$) in the multivariate analysis.

## Progression-Free Survival (PFS)

Median PFS was 70.5 months (IQR [13.5-not reached], and the probability of PFS for 3 years after diagnosis was 53% (Fig 4B). Age at diagnosis, Stage 4, primary tumour excision, bone lysis induced by primary tumour contiguity, SUVpeak, SUVmax, and MTV above 200 cm$^3$ were prognostic factors for PFS in the univariate analysis (Table 3). Patients with an MTV above 200 cm$^3$ had a poorer PFS (3.27 [1.44;7.46]; $p = 0.005$) than patients with an MTV equal to or below 200 cm$^3$ (HR = 3.0; 95% CI [1.5; 6.0]; $p = 0.002$, (Fig 6)). In the multivariate

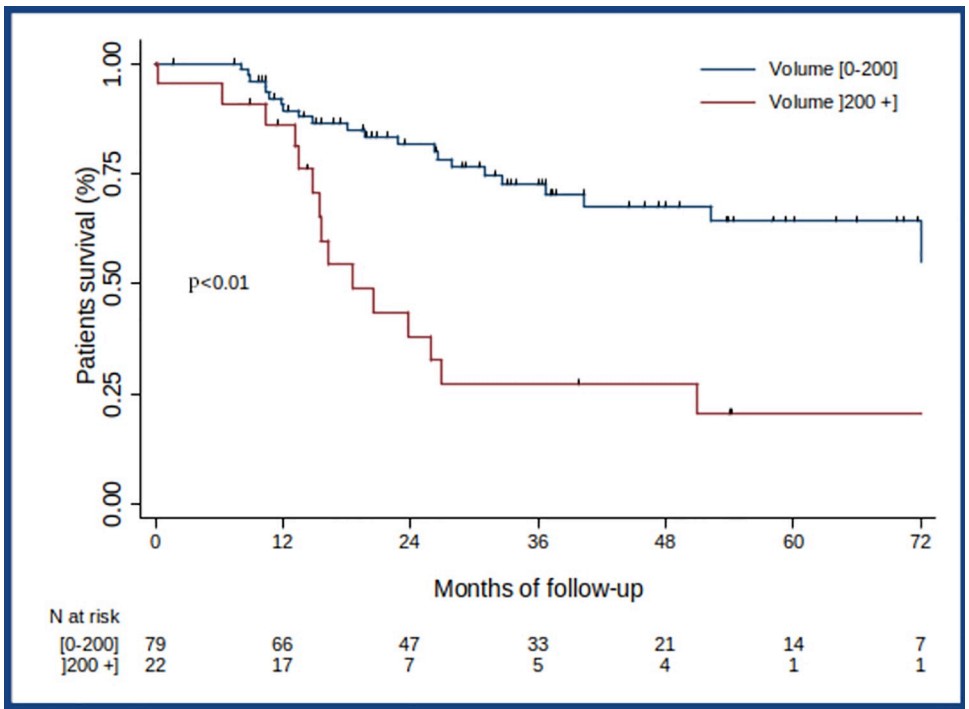

**Fig 5. Overall survival according to MTV.**

analysis, after controlling for confounding effects, the statistically independent clinical or biological factors for PFS that worsened patients' prognosis were Stage 4 ($p = 0.049$), bone lysis induced by primary tumour contiguity ($p = 0.001$), and an MTV above 200 cm$^3$ ($p = 0.006$). Compared with patients with an MTV equal to or below 200 cm$^3$, patients with an MTV above 200 cm$^3$ were 2.7 times more likely to have disease progression (adjusted HR = 2.7; 95% CI [1.3; 5.5] (Table 4)). No statistically independent effect on PFS was found for either SUVpeak ($p = 0.521$) or SUVmax ($p = 0.412$) in the multivariate analysis.

## Discussion

The objective of this 10-year multicentre cohort study was to assess the prognostic value of metabolic tumor volume, measured on PET imaging carried out as part of patients' RMS extension assessment. Thus, after adjusting on confounding factors in the multivariate analysis, the risks of death and recurrence were approximately 2.6 ($p = 0.009$) and 2.7 ($p = 0.006$) times higher for an MTV $\geq$ 200 cm$^3$.

**Table 4. Multivariate analysis, overall & progression-free survival.**

|  | Overall survival HR [95% CI] | *P* value | Progression-free survival HR [95% CI] | *P* value |
|---|---|---|---|---|
| Stage |  |  |  |  |
| 1 | 1 | - | 1 | - |
| 2 | 0.82 [0.15;4.50] | 0.815 | 0.37 [0.07;1.84] | 0.224 |
| 3 | 1.60 [0.513;5.00] | 0.420 | 1.04 [0.40;2.71] | 0.934 |
| 4 | 4.24 [1.336;13.44] | 0.014 | 2.70[1.00;7.28] | 0.049 |
| Bone lysis | 2.91 [1.42;5.94] | 0.003 | 3.16 [1.57;6.35] | 0.001 |
| MTV >200cm3 (primary tumour + distant lesion) | 2.64 [1.27;5.49] | 0.009 | 2.71 [1.32;5.55] | 0.006 |

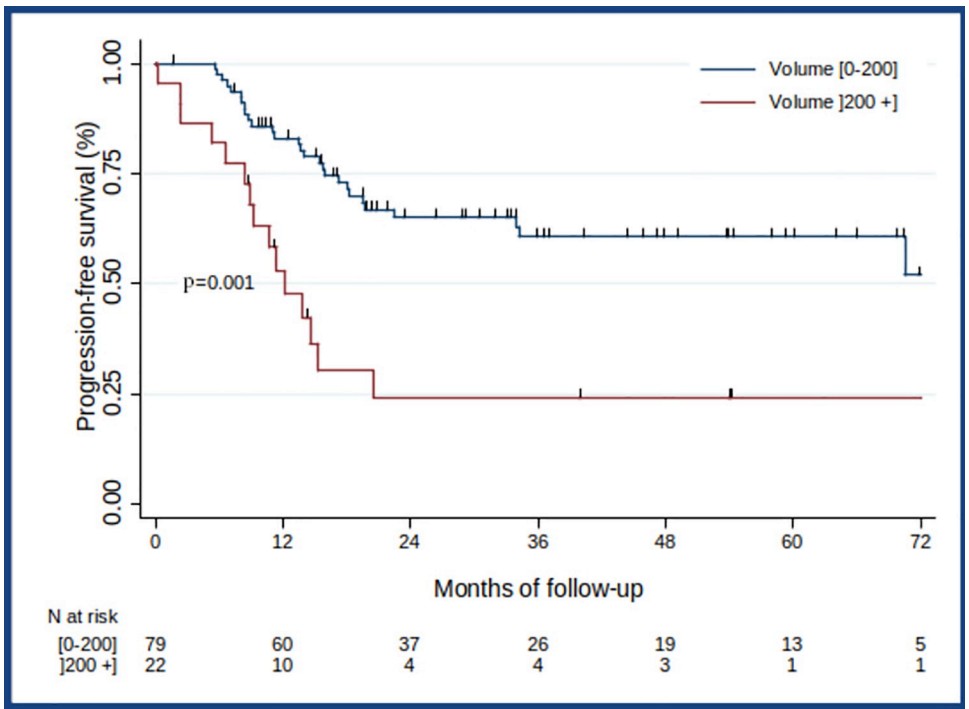

**Fig 6. Progression-free survival according to MTV.**

Although patient selection relied on a PET scan being performed at diagnosis, our population was representative of the clinical reality, especially for the proportion of boys/girls and the distribution in each risk group [26, 38] contrary to previous published PET studies [11, 14, 39]. The age distribution is as previously described with a bimodal age peaks in childhood [40].

Nevertheless, the proportions of alveolar RMS and metastatic RMS were greater in our cohort than in previously reported ones [3, 41, 42]. This could be explained by the exclusion of 16 patients with embryonal RMS at the localized stage, whose primary tumour could not be measured because they had undergone excision surgery before the PET examination was performed. It took a while for PET to become systematically performed at RMS diagnosis, meaning that very few children in the early years of the study underwent PET imaging as part of their extension assessment. thereby 200 patients were also excluded (Fig 1).

The best way of measuring tumour dimensions is a widely debated subject. Although a tumour size of more than 5 cm is historically considered to be a prognostic factor and is used for staging, it seems to be more relevant to measure the tumour in 3D especially for oblong ones. We did not find any publication concerning MTV for RMS in either children or adults, even though this parameter has been reported as a prognostic factor for several other malignant diseases, such as Hodgkin's lymphoma, advanced ovarian squamous cell carcinoma, non-small cell lung cancer, and metastatic colorectal cancer [18–22]. In a previous study of 108 patients with rhabdomyosarcoma combining MRI and CT scans, Ferrari et al. demonstrated a correlation between tumour size and volume with risk of death, with the risk increasingly proportionally to tumour size and volume until a plateau was reached for lesions >12 cm (major axis) or >194 cm³ [43]. However, a correlation between a factor and death cannot be interpreted as a prognostic factor for death. Similarly, Baum et al. showed a correlation between intensity of primary tumour uptake and OS, but failed to demonstrate that it was an

independent predictor of survival [39]. In addition, the use of morphological CT and/or MRI measurements in Ferrari et al.'s study did not take the aggressiveness of the tumour into account, contrary to PET, which provides information about its metabolic activity. These findings are consistent with our results concerning the increased risk of progression and death, linked to MTV increase. In our population, a tumour size > 5cm was taken into account through RMS stage. Thus, after adjusting on confounding factors in the multivariate analysis, including RMS stage, the risks of death and recurrence were approximately 2.6 ($p = 0.009$) and 2.7 ($p = 0.006$) times higher for an MTV $\geq 200$ cm$^3$ ($p = 0.024$).

The advantage of using MTV over tumour size is the possibility of over-classifying small metabolically active lesions as higher-risk lesions. We did not measure the tumour volume from the CT scan, mainly because a large number of children had a chest CT and abdomino-pelvic ultrasound rather than a thoraco-abdominopelvic scan. Some tumours may also have poorly defined contours, and are therefore difficult to measure on the CT, as previously illustrated in Fig 2. Finally, the CT combined with the PET did not include injection with an iodinated contrast medium, so the tumour / anatomical structure contrast was too low to allow for automatic contouring achievable in routine clinical practice.

Instead of exhaustively measuring the volume of each metastasis at diagnosis in the case of polymetastatic disease, we considered a maximum of two lesion volumes (i.e., primary tumour and largest metastasis). This point may be subject to discussion, but we reasoned that it is not reasonable to exhaustively measure the MTV of all metastases in the event of multifocal involvement in clinical daily practice. This type of time-consuming measurement may be appropriate in research protocols, but it is less common in daily use, and may lead to interobserver variations. In the univariate analysis, MTV was associated with a poorer prognosis (both for OS and PFS), even if the volume of the distant lesion was not considered in the MTV calculation. Conceptually speaking, measuring the primary tumour and the largest metastasis appears to be a better approach to gauging the actual tumour burden, and the resulting HR may be closer to reality. This type of approach is already used with the RECIST 1.1 criteria for CT evaluation of the therapeutic response in oncology, involving a maximum of five target lesions, with a maximum of two per organ [44, 45].

Most of PET studies use SUV, and especially SUVmax, to approach the tumor's agressiveness, but with a certain variability of the measure [15, 16]. Thus Brendle et al. showed that SUV calculated on the same PET equipment acquisition was subject to variability according to the differents algorithms reconstructions, and that SUVmax was the least reproducible measurement comparing to SUVmean and SUVpeak [46]. And the meta-analysis of Ghooshkhanei et al. in endometrial cancer illustrates this issue showing that three studies reported an association between the pre-operative SUVmax with disease free survival and/or overall survival. The HR of each study was calculated according to three differents cut off SUVmax (the values were 12.7, 17.7, and 8.35), highliting its variability, even more when PET machines are differents [47].

The prognostic value of the SUVmax of the primary tumour is still equivocal in rhabdomyosarcoma. Baum et al.'s study among 41 patients failed to prove that either primary tumour intensity or SUVmax/SUVliver was an independent predictor of OS and PFS [39]. Neither did Esraa El-Kholy et al.'s more recent study, despite a larger population of 98 patients [14]. By contrast, Casey et al.'s study involving 107 patients showed that a SUVmax threshold of 9.5 for primary tumour was an independent predictor of OS and EFS [11].

We demonstrated that SUVpeak had the same prognostic values across OS and PFS, with an identical HR to SUVmax, but a lower $p$ value. As SUVpeak is calculated by averaging the SUV values in the pixels adjacent to the pixel representing the SUVmax within a radius of 1 cm$^3$, it is free from the vagaries of variation that affect SUVmax, particularly in relation to

background activity, variations in equipment, imaging acquisition and reconstruction protocols, and time between injection and PET acquisition. It therefore seems to be more precise and robust than the SUVmax, as has already been shown in numerous studies [21, 48, 49].

In our study, the SUVmax and SUVpeak values did not appear to be more predictive when adjusted to MTV in the multivariate analysis. This suggests that they act as a confounding factor in the prediction of death or progression, and that it is preferable to only consider MTV.

Among the other factors we evaluated, bone lysis induced by primary tumour contiguity was an independent prognostic factor (Fig 3). Even if the possibility of complete surgical excision of the primary tumour is a known prognostic factor, partly dependent on its locoregional spread and therefore on the involvement of bone invasion [26], we did not find any study that specifically assessed this parameter in childhood RMS. In a clinical data review of 874 adults treated for soft-tissue sarcoma, bone invasion was a prognostic factor [50]. However, none of the 48 patients with bone disease had RMS. In other pathologies such as Hodgkin's or non-Hodgkin's lymphoma, bone invasion of a lymph node in the case of localized disease is not considered to have a worse prognosis and does not lead to any change in disease staging [51]. It might be worthwhile confirming our result by conducting a further prospective study.

We did not find any difference in OS and PFS for histological type or Stages 1, 2 or 3. Only Stage 4 was linked to prognosis. Nowadays, the therapeutic escalation in the RMS 2005 protocol according to the prognostic factors at onset may improve patients' survival. It is only in patients with advanced disease that treatment has not been sufficiently effective, as reported by the Children's Oncology Group Soft-Tissue Sarcoma Committee [52].

The main limiting factor of our study was the population size, despite the recruitment of patients in six major French hospitals and cancer centres. The nationwide collection of patient data would allow us to validate our results in the future. Given the therapeutic challenges in a paediatric population, with the risk of developing secondary toxicities either immediately following treatment (vincristine-induced neuropathies, doxorubicin-induced heart failure) or later on (secondary blood diseases, neoplasia, or post-radiotherapy morphological sequelae), the inclusion of PET parameters such as metabolic volume and SUV in the decision-making trees for the management of RMS could make it possible to adapt patients' therapeutic management.

## Conclusion

This multicentre study, a collaboration of six french university hospital, confirmed the prognostic value of pretreatment PET in childhood RMS. Moreover it is to our knowledge the first time that MTV appears to be a prognostic parameter. By considering the MTV of the primary tumour and the largest distant lesion, where relevant, to gauge the actual metabolic tumour burden, we showed that a MTV $> 200$ cm$^3$ is prognostic on survival with a risk of death or progression multiplied by approximately 2.5. These results should be prospectively validated in a larger patient population. Given the therapeutic challenges in a paediatric population, with the risk of developing secondary toxicities, our study brings an additional argument to include metabolic PET parameters in the decision-making trees for the management of RMS. And it could help to adapt patients' therapeutic management.

Multicenters international studies, especially those of the European paediatric Soft tissue sarcoma Study Group, were focused on the treatment until now. On the future it could be interested to collect methodically patient's PET data then to analyse PET parameters and study their relationship with survival. Thereby the prognostic advantage of measuring MTV should be confirmed by a further prospective multicentre study involving a larger patient population. Thus future protocols could include PET data to classify patients in the different treatment groups and specify the management of children with RMS.

## Supporting information

**S1 Data.**
(XLSX)

## Acknowledgments

The authors thank all the following members who also participated in the trial: F. CACHIN, MD, PhD, J. KANOLD LASTAWIECKA, MD, PhD (Clermont-Ferrand), O. MUNDLER, MD, PhD, A. ROME, MD (Marseille), C. BERGERON, MD, PhD (Lyon). D. MARIANO GOULART MD, PhD (Montpellier), and N. SIRVENT, MD, PhD (Montpellier).

## Author Contributions

**Conceptualization:** Marie-Pierre Castex, Pierre Payoux, Anne Hitzel.

**Formal analysis:** Helio Fayolle, Valerie Lauwers-Cances.

**Investigation:** Anne Hitzel.

**Methodology:** Valerie Lauwers-Cances, Pierre Payoux, Anne Hitzel.

**Resources:** Nina Jehanno, Marie-Pierre Castex, Daniel Orbach, Thomas Mognetti, Corradini Nadège, Pierre Payoux, Anne Hitzel.

**Supervision:** Pierre Payoux.

**Validation:** Pierre Payoux, Anne Hitzel.

**Visualization:** Helio Fayolle, Pierre Payoux, Anne Hitzel.

**Writing – original draft:** Helio Fayolle, Anne Hitzel.

**Writing – review & editing:** Nina Jehanno, Marie-Pierre Castex, Daniel Orbach, Thomas Mognetti, Corradini Nadège, Pierre Payoux, Anne Hitzel.

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
