## [Decision Letter · Decision Letter 0]

8 Oct 2021

PONE-D-21-29606PET metabolic tumor volume as a new prognostic factor in childhood rhabdomyosarcoma.PLOS ONE

Dear Dr. Fayolle,

Thank you for submitting your manuscript to PLOS ONE. After careful consideration, we feel that it has merit but does not fully meet PLOS ONE’s publication criteria as it currently stands. Therefore, we invite you to submit a revised version of the manuscript that addresses the points raised during the review process.

We look forward to receiving your revised manuscript.

Kind regards,

Domenico Albano

Academic Editor

PLOS ONE

Journal Requirements:

Additional Editor Comments:

The paper is relevant and of clinical interest. Only minor points need to be clarified.

Reviewers' comments:

Reviewer's Responses to Questions

**Comments to the Author**

1. Is the manuscript technically sound, and do the data support the conclusions?

Reviewer #1: Yes

Reviewer #2: Yes

2. Has the statistical analysis been performed appropriately and rigorously? 

Reviewer #1: Yes

Reviewer #2: Yes

3. Have the authors made all data underlying the findings in their manuscript fully available?

Reviewer #1: Yes

Reviewer #2: Yes

4. Is the manuscript presented in an intelligible fashion and written in standard English?

Reviewer #1: Yes

Reviewer #2: No

5. Review Comments to the Author

Reviewer #1: This manuscript expand the knowledge on the utility of PET in patients with RMS. I have few indications for enriching the paper.

Please expand and provide examples on conflicting results on the prognostic utility of SUV.

Explain also in your introduction the potential advantages of using MTV instead of SUV.

Explain the choice of 40% threshold for MTV.

Reviewer #2: It is a very interesting study that investigates PET metabolic tumor volume at the time of diagnosis as a new prognostic factor in childhood rhabdomyosarcoma. It is a very useful study, as PET is increasingly used in this tumor and this new marker could be used more widely.

It is the first study investigating MTV in rhabdomyosarcoma and it will be necessary to validate the results in a larger case series.

I have some requests and comments:

There are some wrong punctuation in the paper. Please check and edit.

In the introduction:

-Please add the lung CT scan as part of the initial staging exams

-The paragraph: "identification of unfavorable factors such as age ..." is not clear. Please distinguish the prognosis of localized patients without mentioning the distant metastatic tumor spread, compared to patients with metastatic disease.

-In the sentence "...leading to a cure rate close to 70% in RMS sites": please change "RMS sites" with "localized RMS"

-”PET has been shown to be superior to CT for the initial staging of RMS”. This sentence is not completely true. Some studies (of which the most recent and extensive by Mercolini et al., EJC) demonstrated a superiority of PET compared to CT in some sites but not in others (eg intrathoracic sites). Please edit the sentence and add a reference to the bibliography

-Please use “nodal” instead of “ganglion”

In the “patients population” paragraph:

-The sentence “patients who had undergone a PET-CT after excision of the primary tumor …, were exluded” in my opinion it is not well written and needs to be clarified

In the “Treatment” paragraph:

-The sentence “The monitoring methods after the end of the treatment were also defined by the protocol” is repeated 2 times. Please remove 1 of the two.

In the “Statistical analysis” paragraph:

- please remove the first sentence ("Because RMS is a rare disease, several centers .... sample size population")

In the discussion:

- “The age distribution is as reported as previous study with a peak around young infancy”. The reference 29 is not a study and in my opinion the sentence is not clear. Pleas edit.

- "in a previous study of 108 patients combining MRI and CT scans, Ferrari et al .....". Please specify that these are patients with rhabdomyosarcoma.

6. PLOS authors have the option to publish the peer review history of their article (what does this mean?). If published, this will include your full peer review and any attached files.

Reviewer #2: No

---

## [Author Response · Author response to Decision Letter 0]

27 Oct 2021

Journal Requirements :

We tried to do it well, following Plos ONE’s instructions.

According to the french law, the retrospectives non interventional studies do not require patient consent when the study protocol is compliant with the CNIL reference methodology repository about the retrospective data collection. In the case of an expressly written refusal, patient data were not analysed. This has allowed our work to be approved by the Ethics Committee of French Society of Nuclear Medicine and registered with the number CEMEN 2020-01. Thus, we could analyse PET images from patients and provide illustrative images.

Not concerned.

Not concerned.

Not concerned.

It is now added in the cover letter.

The data set is named data_set, in two files, Access and Excel. We removed the birth dates in order to not compromise confidentiality in the context of human-subject research. Date of diagnosis, relapse, death, still remain.

It is ok.

It is ok.

Reviewers :

#1 :

1. Please expand and provide examples on conflicting results on the prognostic utility of SUV.

Added in the discussion :

Most of PET studies use SUV, and especially SUVmax, to approach the tumor’s agressiveness, but with a certain variability of the measure (15,16). Thus Brendle et al. showed that SUV calculated on the same PET equipment acquisition was subject to variability according to the differents algorithms reconstructions, and that SUVmax was the least reproducible measurement comparing to SUVmean and SUVpeak (46). And the meta-analysis of Ghooshkhanei et al. in endometrial cancer illustrates this issue showing that three studies reported an association between the pre-operative SUVmax with disease free survival and/or overall survival. The HR of each study was calculated according to three differents cut off SUVmax (the values were 12.7, 17.7, and 8.35), highliting its variability, even more when PET machines are differents (47).

2. Explain also in your introduction the potential advantages of using MTV instead of SUV.

This was added in the introduction complementary to the shorter justification of MTV choice already written:

Single pixel values of the SUV, and especially SUVmax, are commonly used as a quantitative index of tumor metabolism, mainly because it is now well implemented on images viewers and thus easy to use. However this semiquantitative evaluation is subject to intra- and interindividual biases by a broad range of biological and technical factors such as patient’s weight, blood glucose level, acquisition parameters including uptake time, inaccurate calibration of PET, image reconstruction algorithm, etc. (15,16). To overcome these shortcomings, MTV approach, defined as the sum of the volume of voxels with SUV surpassing a threshold value in a tumor, can be considered (17). Recent studies confirmed the interest of MTV and sometimes its superiority compared to SUVmax with regard to prognostic value, in various neoplastic pathologies such as Hodgkin's lymphoma, ovarian squamous cell carcinoma, non-small cell lung cancer, metastatic colorectal cancer, and pancreatic cancer (18–23).

3. Explain the choice of 40% threshold for MTV.

This was added in the Methods and Materials part in PET protocol, complementary to the shorter justification already written: 

From the physics side, after phantom studies (28,29), a threshold value of 40% to define the tumor boundary on PET images was used in many clinical studies (30–33). This 40% threshold is the most common index in clinical practice for evaluating tumor prognosis (34,35). Nowadays PET imaging softwares offer an automatic 40% SUV approach to delineate tumor contours. Thereby a threshold of 40% of the SUVmax was applied in our study.

#2 :

There are some wrong punctuation in the paper. Please check and edit.

1-In the introduction:

-Please add the lung CT scan as part of the initial staging exams.

The lung CT scan as a part of the initial staging exams was added:

“After initial clinical symptoms, such as a swelling, paraclinical tests [...] lesion biopsy, local imaging (lung-CT, MRI), lumbar puncture […] metastatic lesions (5).”

-The paragraph: "identification of unfavorable factors such as age ..." is not clear. Please distinguish the prognosis of localized patients without mentioning the distant metastatic tumor spread, compared to patients with metastatic disease.

-In the sentence "...leading to a cure rate close to 70% in RMS sites": please change "RMS sites" with "localized RMS"

The manuscript has been modified :

Indeed, the global cure rate for RMS (for all risks groups) has improved from 25–30% to approximately 70%, in the 30 last years.

“Identification of unfavourable factors such as age (> 10 years), alveolar histological subtype, size (> 5 cm) and location of the primary tumour (parameningeal, limbs, or trunk), and presence of regional nodal or distant metastatic tumour spread sites, has allowed a risk classification to optimize the treatment the combination of chemotherapy and surgery to be optimized, leading to an improvement of the cure rate from 25–30% to approximately 70% (6). Thus patients in the low risk group have the best prognosis progression-free survival (PFS) and overall survival (OS) (3-year PFS rate of 88%) (7). However, patients with metastatic disease still have a dismal prognosis (OS: 0‑30%) (8)..”

-”PET has been shown to be superior to CT for the initial staging of RMS”. This sentence is not completely true. Some studies (of which the most recent and extensive by Mercolini et al., EJC) demonstrated a superiority of PET compared to CT in some sites but not in others (eg intrathoracic sites). Please edit the sentence and add a reference to the bibliography

The sentence has been modified :

“Apart from detecting intrathoracic lesions where chest CT remains essential, F-18 FDG PET-CT (PET) has been shown to be superior to CT for the initial staging of RMS, mainly on account of its ability to detect nodal involvement and metastatic disease (9)”

-Please use “nodal” instead of “ganglion”

The term has been changed.

2- In the “patients population” paragraph:

-The sentence “patients who had undergone a PET-CT after excision of the primary tumor …, were exluded” in my opinion it is not well written and needs to be clarified

The sentence has been modified :

“Patients who had undergone a PET-CT examination while the primary tumor was already excised by surgery, who had commenced chemotherapy, who had RMS located in the bladder or in a parameningeal site (where measuring MTV was impossible, owing to the close physiological activity of the bladder or brain), whom PET images where unrecorded, or who had a tumour in a limb that was not within the scope of acquisition, were excluded.”

3- In the “Treatment” paragraph:

-The sentence “The monitoring methods after the end of the treatment were also defined by the protocol” is repeated 2 times. Please remove 1 of the two.

The repeated sentence is deleted.

4- In the “Statistical analysis” paragraph:

- please remove the first sentence ("Because RMS is a rare disease, several centers .... sample size population")

The first sentence was deleted

5- In the discussion:

- “The age distribution is as reported as previous study with a peak around young infancy”. The reference 29 is not a study and in my opinion the sentence is not clear.

The sentence and the reference have been modified :

“The age distribution is as previously described with a bimodal age peaks in childhood (40).”

- Please edit.

"in a previous study of 108 patients combining MRI and CT scans, Ferrari et al .....". Please specify that these are patients with rhabdomyosarcoma.

It is added.

---

## [Decision Letter · Decision Letter 1]

6 Dec 2021

PET metabolic tumor volume as a new prognostic factor in childhood rhabdomyosarcoma.

PONE-D-21-29606R1

Dear Dr. Helio FAYOLLE,

We’re pleased to inform you that your manuscript has been judged scientifically suitable for publication and will be formally accepted for publication once it meets all outstanding technical requirements.

Kind regards,

Domenico Albano

Academic Editor

PLOS ONE

Additional Editor Comments (optional):

Reviewers' comments:

Reviewer's Responses to Questions

**Comments to the Author**

1. If the authors have adequately addressed your comments raised in a previous round of review and you feel that this manuscript is now acceptable for publication, you may indicate that here to bypass the “Comments to the Author” section, enter your conflict of interest statement in the “Confidential to Editor” section, and submit your "Accept" recommendation.

Reviewer #1: All comments have been addressed

Reviewer #2: All comments have been addressed

2. Is the manuscript technically sound, and do the data support the conclusions?

Reviewer #1: Yes

Reviewer #2: Yes

3. Has the statistical analysis been performed appropriately and rigorously? 

Reviewer #1: Yes

Reviewer #2: Yes

4. Have the authors made all data underlying the findings in their manuscript fully available?

Reviewer #1: Yes

Reviewer #2: Yes

5. Is the manuscript presented in an intelligible fashion and written in standard English?

Reviewer #1: Yes

Reviewer #2: Yes

6. Review Comments to the Author

Reviewer #1: I do not have further comments. I am very happy with the edits presented in the new version since the quality has improved.

Reviewer #2: All comments and suggestions has been satisfied. The paper is certainly very interesting and deserves to be published

7. PLOS authors have the option to publish the peer review history of their article (what does this mean?). If published, this will include your full peer review and any attached files.

Reviewer #1: **Yes: **Natale Quartuccio

Reviewer #2: No

---

## [Editor Report · Acceptance letter]

6 Jan 2022

PONE-D-21-29606R1 

PET metabolic tumor volume as a new prognostic factor in childhood rhabdomyosarcoma. 

Dear Dr. FAYOLLE:

I'm pleased to inform you that your manuscript has been deemed suitable for publication in PLOS ONE. Congratulations! Your manuscript is now with our production department. 

Kind regards, 

on behalf of

Dr. Domenico Albano 

Academic Editor

PLOS ONE